# Dynamic earthquake rupture preserved in a creeping serpentinite shear zone

Matthew S. Tarling [1], Steven A.F. Smith [1], Cecilia Viti [2] & James M. Scott [1]

Laboratory experiments on serpentinite suggest that extreme dynamic weakening at earthquake slip rates is accompanied by amorphisation, dehydration and possible melting. However, hypotheses arising from experiments remain untested in nature, because earthquake ruptures have not previously been recognised in serpentinite shear zones. Here we document the progressive formation of high-temperature reaction products that formed by coseismic amorphisation and dehydration in a plate boundary-scale serpentinite shear zone. The highest-temperature products are aggregates of nanocrystalline olivine and enstatite, indicating minimum peak coseismic temperatures of ca. $925 \pm 60\,°C$. Modelling suggests that frictional heating during earthquakes of magnitude 2.7–4 can satisfy the petrological constraints on the coseismic temperature profile, assuming that coseismic fluid storage capacity and permeability are increased by the development of reaction-enhanced porosity. Our results indicate that earthquake ruptures can propagate through serpentinite shear zones, and that the signatures of transient frictional heating can be preserved in the fault rock record.

[1] Department of Geology, University of Otago, 360 Leith Street, 9016 Dunedin, New Zealand. [2] Università degli Studi di Siena, Dipartimento di Scienze Fisiche, della Terra e dell'Ambiente, Via Laterino 8, Siena 53100, Italy. Correspondence and requests for materials should be addressed to M.S.T. (email: tarma723@student.otago.ac.nz)

Serpentinite is an important rock type in a range of tectonic settings[1,2], including the slab-mantle interface in subduction zones[3–7], oceanic detachment faults[8–10] and large-displacement transform faults in the oceanic and continental lithosphere[11–14]. Where serpentinite shear zones can be observed at the surface or in drill-cores, they often show a pervasive foliation and contain microstructural evidence to suggest that distributed deformation occurs by pressure-solution processes during fault creep[15,16]. Combined with the results of laboratory friction experiments, these characteristics have been used to argue that serpentine, together with common accessory minerals, such as talc, may account for the creeping behaviour of some fault zones[17–19].

The results of numerical modelling[20], geophysical observations[21,22] and laboratory experiments[23,24] show that a transition from stable creep at low slip rates to unstable rupture at high slip rates is possible on the same fault patch. In the case of serpentinite, it has been proposed that this transition in fault stability is enhanced at high slip rates by dynamic weakening mechanisms that involve flash heating of asperity contacts[25–27], thermal pressurisation of pore fluids[28–30] and mineral dehydration reactions that release structurally-bound fluid into the coseismic slip zone[31,32]. However, ancient earthquake ruptures have not previously been recognised in exhumed serpentinite shear zones[33] and thus hypotheses developed from deformation experiments and numerical modelling concerning possible dynamic weakening mechanisms remain untested in nature. Whether creeping serpentinite shear zones can transiently host dynamic earthquake ruptures remains an open question.

## Results

**Structure of the Livingstone Fault, New Zealand.** The Livingstone Fault in New Zealand is a plate boundary-scale serpentinite shear zone that separates ultramafic rocks of the Dun Mountain Ophiolite Belt from metasediments of the continental Caples Terrane[34–36]. The shear zone is dominated by a serpentinite mélange tens to several hundreds of metres wide (Fig. 1a) in which a pervasive anastomosing fabric is well developed (Fig. 1b). The presence of this fabric, together with the abundance of chrysotile veins and the widespread occurrence of dissolution surfaces enriched in second phases (e.g., magnetite), suggest that distributed deformation involved pressure solution[37]. Embedded within the shear zone are pods of more competent material (e.g., metasediments, rodingite, massive serpentinite) ranging from tens to hundreds of metres in size (Fig. 1c). Polished and striated fault surfaces cut across the main shear zone fabrics and are also commonly found along the margins of the pods (Fig. 1d). The fault surfaces separate foliated serpentinite from layers of magnetite ~100 to 300 μm thick (Fig. 2a–c) and probably formed due to the mechanical contrast provided by the bimaterial serpentinite–magnetite interface. Elongate inclusions of serpentinite up to 1000 μm-long and 50 μm-thick are found inside the magnetite layers (Fig. 2b, c). We analysed the content of these serpentinite inclusions by transmission electron microscopy (TEM) and document below the appearance of distinct textures and mineral assemblages that can be related to the progressive amorphisation and dehydration of serpentinite within the inclusions, which we interpret to have occurred during propagation of a coseismic thermal pulse.

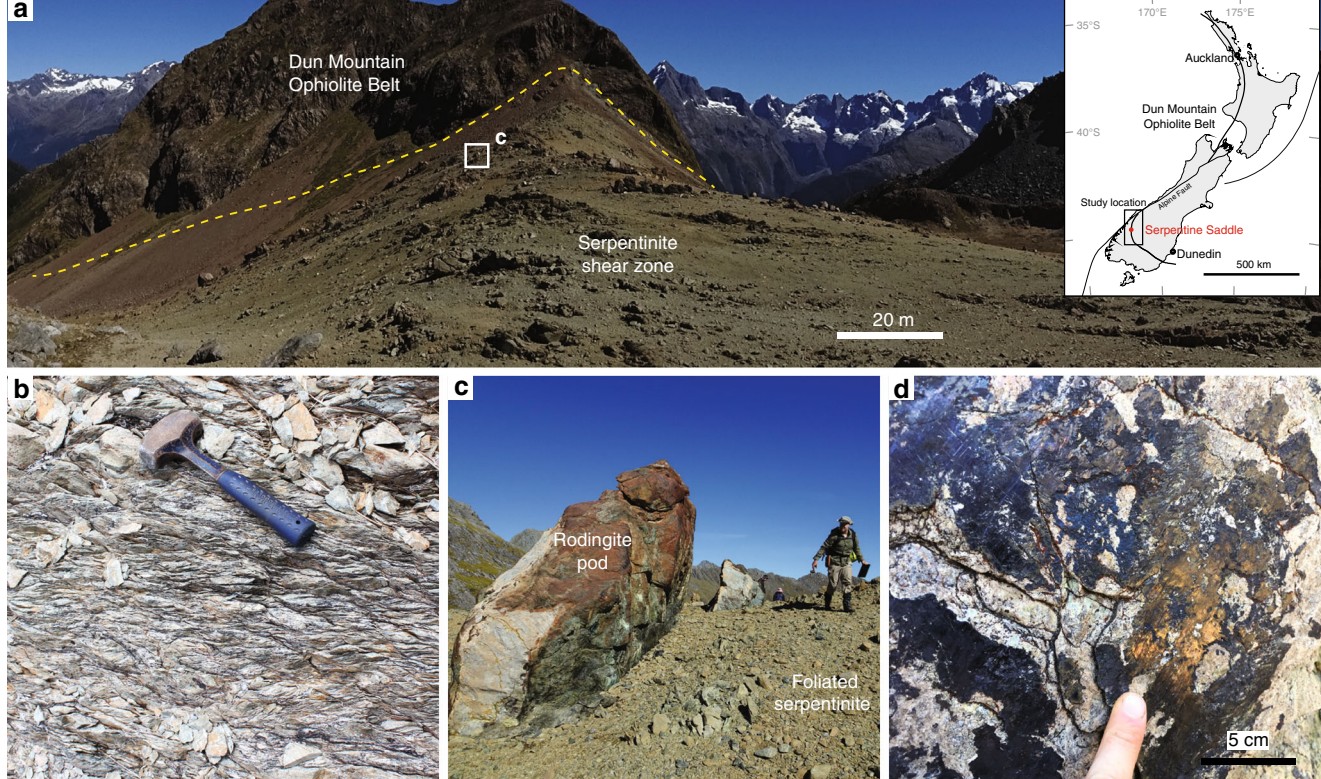

**Fig. 1** The Livingstone Fault serpentinite shear zone. **a** Overview of a ~400 m-wide section of the foliated serpentinite shear zone exposed at Serpentine Saddle (−44.65149, 168.16577). The inset shows the segment of the Livingstone Fault studied in this work and the specific field location from which the photo in part (**a**) was taken. **b** Foliated serpentinite with a pervasive anastomosing fabric. Each lens-shaped domain of serpentinite is coated by fibrous chrysotile. **c** Pod of rodingite surrounded by foliated serpentinite. **d** Striated and polished fault surface coated by a 300 μm-thick layer of magnetite

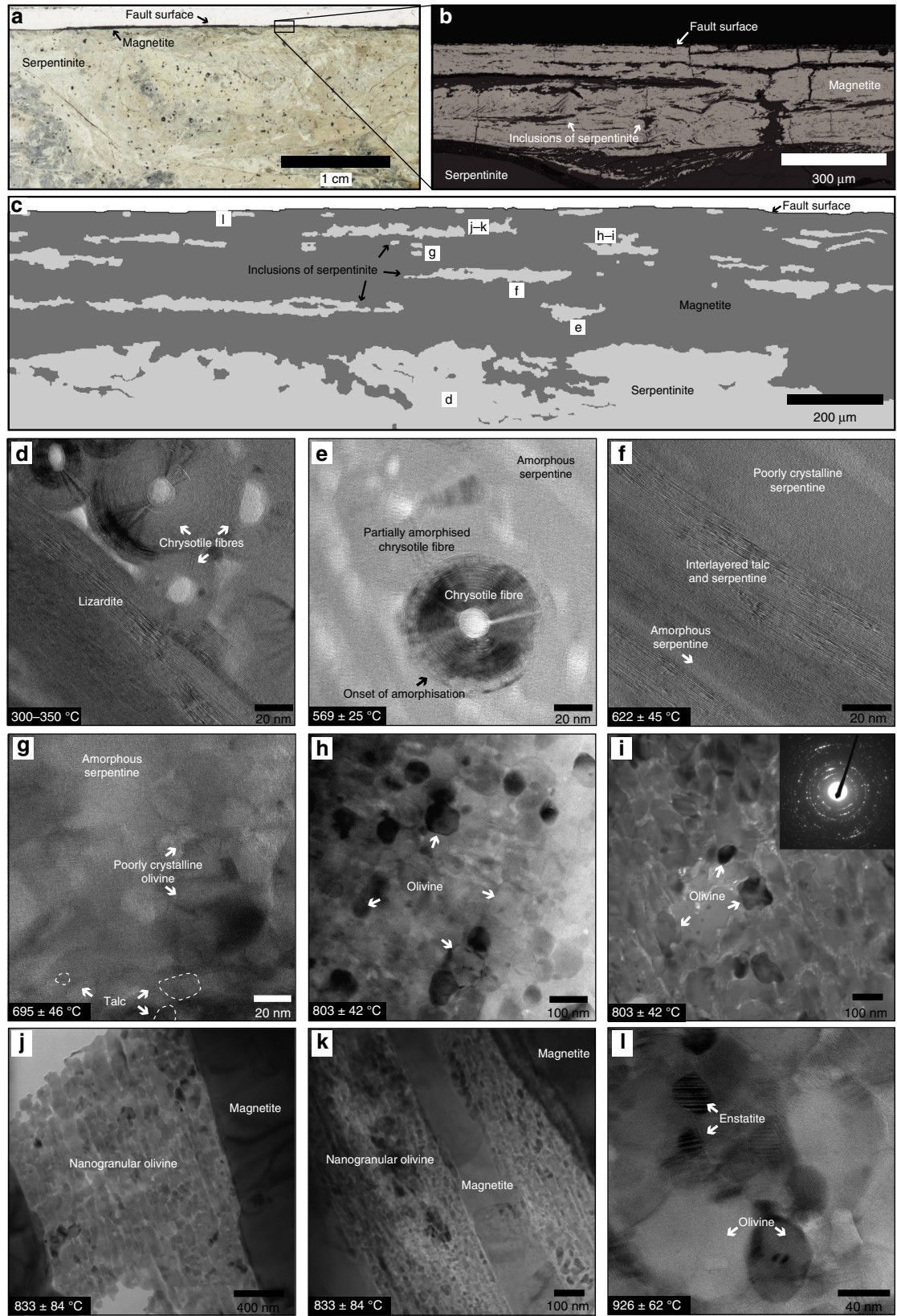

**Petrology and microstructures of the serpentinite inclusions**.
Outside the magnetite layers, the foliated serpentinite shear zone
consists entirely of crystalline lizardite and fibrous chrysotile
(Fig. 2d), with no relict olivine or pyroxene, consistent with the
estimated ambient temperature during shearing of 300–350 °C (see
Methods and Supplementary Table 1 for details of all of the tem-
perature estimates). The inclusions that are furthest from the
polished fault surfaces show the onset of serpentine amorphisation
(Fig. 2e). Remnants of partially amorphised chrysotile fibres and
layers of lizardite exist in an amorphous serpentine matrix (Fig. 2e),

**Fig. 2** Magnetite fault surfaces and serpentine dehydration products. **a** Petrographic thin section of a polished fault surface, thin magnetite layer, and adjacent foliated serpentinite. The foliated serpentinite on the other side of the fault surface was not preserved. **b** Scanning electron microscope image of the polished fault surface, magnetite layer, and elongate inclusions of serpentinite within the magnetite layer. **c** Schematic representation of the magnetite layer with inclusions of serpentinite. Inclusions with letters correspond to the transmission electron microscope (TEM) images shown below. **d** Fibrous chrysotile and lizardite. **e** Partially amorphous chrysotile fibres in a matrix of amorphous serpentine. **f** Poorly crystalline to amorphous serpentine intergrown with talc. **g** Incipient olivine and talc in a matrix of amorphous serpentine. **h, i** Moderately crystalline olivine. Inset in **i** shows selected area electron diffraction pattern (SAED) for nanogranular olivine. **j, k** Aggregates of well-crystallised nanogranular olivine surrounded by magnetite. Olivine nanograins are typically elongate in the slip direction. **l** Crystalline nanograins of olivine and enstatite. All temperature estimates are detailed in the text and listed in Supplementary Table 1

estimated to form at ~569 ± 25 °C. Closely associated with the onset of serpentine amorphisation, thin lamellae of talc are observed to grow intimately with serpentine at ~622 ± 45 °C (Fig. 2f). Towards the fault surface, the occurrence of poorly crystalline nanograins of olivine coexisting with talc and amorphous material, with little to no crystalline serpentine, indicates formation at ~695 ± 46 °C (Fig. 2g). At distances of less than ca. 100 μm from the fault surface, talc is no longer present and the assemblage consists of aggregates of moderately crystalline olivine (20–100 nm grain size) with minor amounts of amorphous material (Fig. 2h, i). The disappearance of talc is estimated to occur at ~803 ± 60 °C. Closer still to the fault surface, the inclusions contain aggregates of well-crystallised nanogranular olivine with negligible amorphous material, consistent with ~833 ± 84 °C (Fig. 2j, k). The nanograins range from 50 to 200 nm in size and are aligned with their grain long axes sub-parallel to the fault slip direction. The inclusions immediately adjacent to the fault surface consist entirely of nanogranular enstatite and olivine with well-defined grain boundaries, euhedral to subhedral grain shapes and negligible porosity, suggesting formation at ~926 ± 62 °C (Fig. 2l). The final constraint is that the magnetite (or any other materials present) shows no indication of having experienced melting, which is predicted to occur at a temperature of 1597 °C[38].

**Amorphisation and dehydration of serpentinite**. The microstructures and mineral assemblages preserved in the serpentinite inclusions (Fig. 2) are strikingly similar to those produced in high-velocity friction experiments on serpentinite[39,40] and are consistent with a reaction sequence involving progressive dehydration of serpentinite to form poorly crystalline or amorphous serpentine, talc, forsteritic olivine and then enstatite[41,42]. At the onset of dehydration at relatively low temperatures (~500–600 °C), the reactions can be described by the formation of talc and forsterite from serpentine:

$$5\,\text{Serpentine} \rightarrow \text{Talc} + 6\,\text{Forsterite} + 9\text{H}_2\text{O}$$
$$5\text{Mg}_3\text{Si}_2\text{O}_5(\text{OH})_4 \rightarrow \text{Mg}_3\text{Si}_4\text{O}_{10}(\text{OH})_2 + 6\text{Mg}_2\text{SiO}_4 + 9\text{H}_2\text{O} \tag{1}$$

At higher (> 800 °C) temperatures, the reaction is best represented by the complete dehydration of serpentine to forsterite and enstatite:

$$\text{Serpentine} \rightarrow \text{Forsterite} + \text{Enstatite} + 2\text{H}_2\text{O}$$
$$\text{Mg}_3\text{Si}_2\text{O}_5(\text{OH})_4 \rightarrow \text{Mg}_2\text{SiO}_4 + \text{MgSiO}_3 + 2\text{H}_2\text{O} \tag{2}$$

It is important to note that the temperature estimates provided above (and outlined fully in the Methods and Supplementary Table 1) are derived from a review of the published literature on static heating experiments, in which dehydration reactions occurred in thermodynamic equilibrium. Therefore, the temperature estimates are likely to represent lower bounds on

possible coseismic reaction temperatures, because during coseismic slip the reaction boundaries may have been significantly overstepped and were more likely driven by kinetics than equilibrium.

**Modelling coseismic frictional heating**. To test whether a coseismic thermal pulse could be responsible for the observed dehydration sequence, finite-element modelling was used to quantify the effects of frictional heating for a range of earthquake magnitudes. The mathematical framework and governing equations used in the numerical model are presented in the Methods section. When frictional heating occurs within a serpentinite-bearing fault, a number of coupled physical and chemical processes govern the evolution of heat production and transfer. Frictional heating leads to the expansion of pore fluids and an increase in pore fluid pressure, which reduces the effective normal stress and leads to thermal pressurisation[28–30]. In addition, thermally driven dehydration and dehydroxylation of serpentine results in significant fluid production (~12–13 wt.% water), which contributes to the thermal pressurisation effect[31]. The reaction of serpentine to olivine has a negative solid volume change, producing up to ~24% porosity[43,44]. The reaction is also endothermic[45], thus consuming some of the energy from frictional heating[46]. To address these coupled processes in our numerical model, we applied the mathematical framework for thermal pressurisation presented by Rice[30], together with the treatment of coseismic dehydration, fluid production and volume change outlined in Brantut et al.[31]. The Brantut et al.[31] study acknowledges the potential importance of reaction-enhanced porosity in controlling fluid pressure evolution during coseismic slip, but it does not explicitly include those effects, instead specifying that the change in porosity can be accounted for by reducing the effective fluid production. Here, we include the effects of reaction-enhanced porosity, which leads to an increase in fluid storage capacity, as well as an increase in permeability within the dehydrating layer during coseismic slip. We model this effect by allowing permeability to increase by up to one order of magnitude with the progress of dehydration within the reaction zone, consistent with the experimental data of Tenthorey and Cox[43]. The initial permeability is set to $10^{-19}$ m$^2$, consistent with laboratory data on the permeability of foliated serpentinite[43,47]. This is allowed to evolve towards a permeability of $10^{-18}$ m$^2$ during coseismic dehydration as a function of the dehydration reaction progress.

The model consists of a 300 μm-thick layer of magnetite sandwiched between layers of serpentinite. A frictional heat source is defined at one of the contacts between the magnetite and serpentinite, corresponding to the bimaterial fault surface observed in the natural samples. The model is evaluated considering the one-dimensional transfer of heat perpendicular to the fault surface. The frictional heat flux from the fault surface is calculated as[48]:

$$Q_{\text{fric}} = \mu(\sigma_{\text{N}} - p_{\text{f}})v \tag{3}$$

where $\mu$ is the sliding friction coefficient, $\sigma_N$ is the normal stress on the fault, $p_f$ is the pore fluid pressure and $\nu$ is the slip velocity.

The model assumes a peak slip velocity of 1 m/s[29,30,49] with a 'boxcar' slip velocity function[50]. Normal stress was set at 270 MPa and the initial pore pressure was 0.3 $\sigma_N$. Following Noda and Shimamoto[51], Griffith et al.[52] and Brantut et al.[31], the sliding friction coefficient is assumed to be 0.4. In nature, it is likely that the friction coefficient drops dramatically during coseismic slip due to dynamic weakening processes, as suggested by theoretical[30,53] and experimental studies[24]. In our model, dynamic weakening is approximated by the reduction in the effective normal stress due to fluid pressurisation, and as such the model captures some of the complexity that likely occurs during weakening processes in nature. As we lack constraints on the width of the coseismic slip zone, we assume a zero-width fault. This may result in an overestimation of heat production at the fault surface for a given slip distance, meaning that our estimates of the slip distances and corresponding earthquake magnitudes required to produce a given thermal profile should be considered as lower bounds. Slip distances and the corresponding range of earthquake magnitudes were taken from the summary of Sibson[54].

During coseismic frictional heating, we evaluate the diffusion of heat and fluids within the fault zone. This is done through the coupling of two main equations: the energy equation and the fluid mass conservation equation (eqs. 4 and 10). The model's energy equation takes the frictional heat flux from eq. 3 and accounts for the energy consumed by dehydration of serpentine[45]. Pore fluid pressure is calculated through the fluid mass conservation equation (eq. 10), which takes into account the pressurisation of the fluid by heating, fluid flow away from the fault, poroelasticity and the fluids and porosity produced by the dehydration reaction. The energy equation is coupled to the fluid mass conservation equation through parameter $p_f$ in eq. 1, as increased pore fluid pressure reduces overall frictional heating.

The result of thermal pressurisation occurring concurrently with the fluid-producing endothermic dehydration reaction is a much lower temperature rise for a given earthquake magnitude compared with a situation in which thermal pressurisation does not occur. However, frictional heating at the slip surface is not completely inhibited by thermal pressurisation, largely due to the generation of substantial reaction-enhanced porosity and the coupled increase in permeability. We note that a one order-of-magnitude increase in permeability due to reaction-enhanced porosity is probably quite conservative, because the extreme stress and temperature conditions associated with rupture propagation are likely to induce coseismic fracturing of the wall rocks[55–57].

## Discussion

In our model, the bimaterial nature of the fault interface leads to the propagation of an asymmetric thermal pulse (Fig. 3). This asymmetry is due to the relatively high thermal conductivity of the magnetite layer compared to the adjacent serpentinite, which allows for a more significant rate of heat conduction away from the fault surface on the magnetite side. Our modelling results suggest that the dehydration products preserved within the serpentinite inclusions could have formed due to propagation of a heat pulse generated by frictional sliding along the fault interface (Fig. 3). We suggest that the high-temperature dehydration assemblages are preserved within the inclusions because they are surrounded on all sides by magnetite, and were therefore protected from hydration and conversion back in to serpentine following coseismic slip. Although the foliated serpentinite on the other side of the fault surface is also predicted to dehydrate during coseismic slip (Fig. 3), any dehydration assemblages in this

region would rapidly be converted back in to serpentinite and become unrecognisable. For the model parameters used here, the petrological constraints on the coseismic thermal profile can be satisfied for earthquakes of approximately $M_w = 2.7–4.0$, corresponding to coseismic displacements in the range of ~0.01–0.03 m (Fig. 3). Instead, if no increase in coseismic porosity and permeability is inferred, thermal pressurisation shuts down any significant frictional heating to the extent that the dehydration reactions are not predicted to occur.

As mentioned previously, the assumption of a zero-width fault, combined with the fact that the dehydration reaction boundaries may have been significantly overstepped during coseismic slip, means that our estimates could represent lower bounds for the earthquake magnitudes responsible for frictional heating. At the same time, there is currently a very limited understanding of coseismic permeability. If the permeability was increased by several orders-of-magnitude by coseismic fracturing of the wall rocks, the petrological constraints can be met by earthquakes of smaller magnitude.

Based on field and microstructural evidence, we interpret that distributed deformation in the shear zone occurred by pressure-solution creep, probably accompanied by crystal plasticity and frictional sliding between serpentine lamellae[15,39,58]. Due to the low-viscosity and ductile nature of serpentine at moderate to high temperatures, combined with its complex frictional behaviour and velocity dependence[19,59], it has previously been suggested that serpentinite shear zones could inhibit the nucleation and propagation of earthquake ruptures[60,61]. However, our new microstructural and petrological observations, taken together with the results of high-velocity rock friction experiments[25–27,32], indicate that dynamic seismic slip can occur within creeping serpentinite shear zones, and that the signatures of coseismic frictional heating can be preserved in the fault rock record.

Field observations from the Livingstone Fault and other large tectonic faults suggest that earthquake nucleation could have occurred within wall rocks adjacent to the shear zone, or within large pods of competent rock embedded within creeping segments (Fig. 1c). Stress concentrations within and along the margins of such pods can lead to brittle failure despite distributed deformation in the shear zone occurring by creep[62]. In addition, interactions between pods could potentially lead to 'log-jams', locking up portions of the shear zone and allowing high stresses to accumulate, ultimately leading to brittle failure[63]. Alternatively, serpentinite shear zones typically contain non-negligible amounts of magnetite (up to ca. 15 wt.%), which is a common product of the primary serpentinisation reaction. Although magnetite is initially disseminated in serpentinite shear zones, it can become concentrated along fault and foliation surfaces by cataclasis, precipitation in veins, or pressure solution, leading to the formation of continuous and interconnected layers of magnetite (Fig. 2a–c). Although extremely thin (typically < 1 mm), these magnetite layers are strong and brittle compared with the serpentinite, and provide a significant mechanical contrast along which unstable slip and earthquake rupture could occur.

## Methods

**Transmission electron microscopy.** Polished petrographic thin sections of the samples were prepared using Canada balsam adhesive. TEM grids were extracted from the sections, mounted on a 3 mm Cu grid with a central 800 μm hole. Samples were milled to electron transparency by Ar$^+$ ion milling (Gatan Dual Ion Mill). The TEM characterisation was performed on a JEOL JEM-2010 microscope, working at 200 kV, with LaB$_6$ source, ultra-high resolution pole pieces, resulting in a 0.19 nm point resolution.

**Numerical model.** Numerical modelling was performed using the software package COMSOL Multiphysics. The model of coseismic dehydration of serpentinite is based on the mathematical framework of thermal pressurisation and coseismic

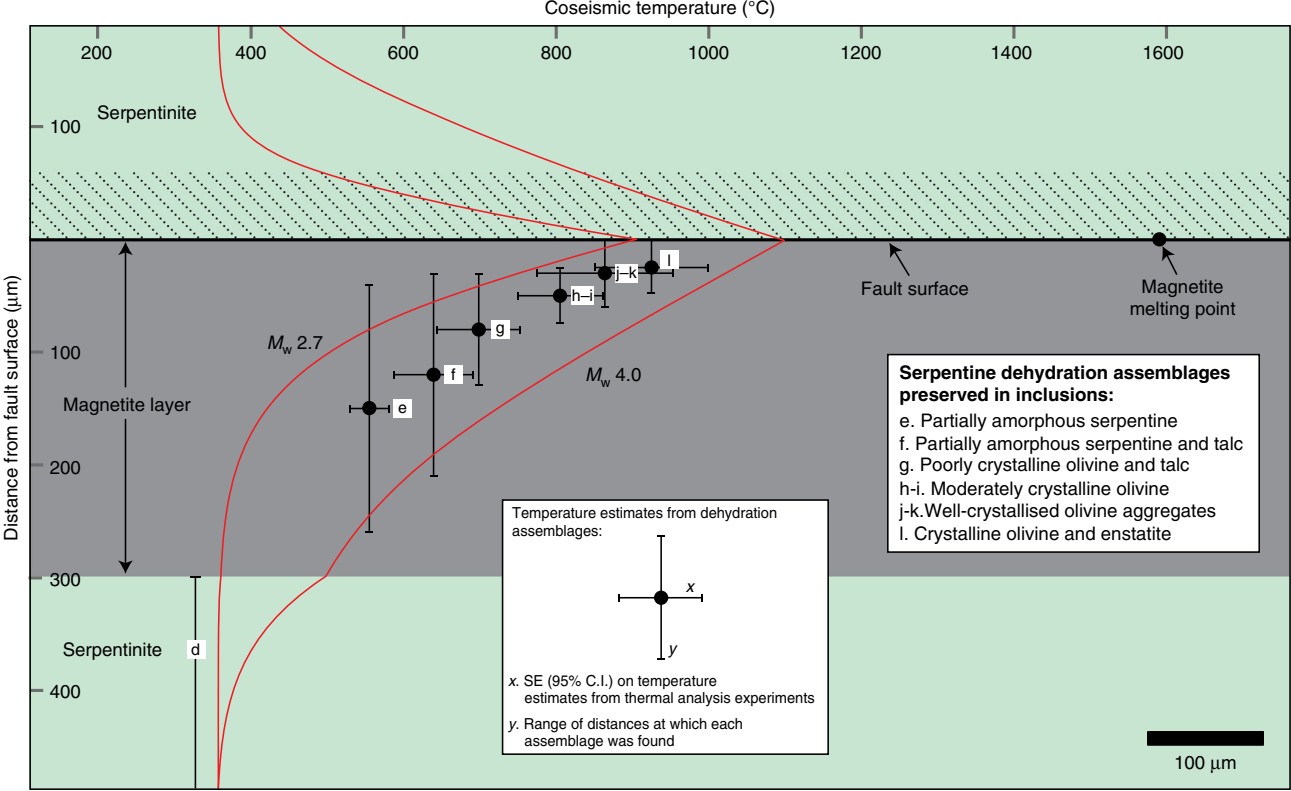

**Fig. 3** Results of finite-element modelling and summary of temperature estimates for dehydration products. Points lettered (e) to (l) represent the locations of serpentinite inclusions and the temperature estimates for the dehydration assemblages found within each inclusion (shown in Fig. 2). Region (d) is the estimated ambient temperature (350 °C) in the foliated serpentinite shear zone (see Methods). X error bars represent the SE (with a 95% confidence interval) of the mean temperature reported for each dehydration assemblage in the literature (see Supplementary Table 1). Y bars represent the range of distances over which each dehydration assemblage was found during TEM examination. The melting point of magnetite is also indicated. The red curves show the modelled temperature pulse resulting from frictional heating during earthquakes of $M_w$ 2.7 and $M_w$ 4.0, which are compatible with the petrological constraints on the dehydration sequence. The hatched area shows the layer of serpentinite that is predicted to dehydrate for a $M_w$ 2.7 earthquake, although the dehydration products in this layer would rapidly be converted back in to serpentine. The same layer would be ca. 160 µm thick for a $M_w$ 4 earthquake

dehydration outlined by Rice[30] and Brantut et al.[31]. The model includes: one-dimensional (1D) heat diffusion during coseismic frictional heating, enthalpy of dehydration of serpentinite, thermal pressurisation of pore fluids in a poroelastic medium, production of fluid during the dehydration of serpentinite and resulting thermochemical pressurisation, reaction-enhanced porosity change and increase in permeability due to reaction-induced porosity. For a complete treatment of the governing equations and their derivation see Lachenbruch[28], Rempel and Rice[29], Rice[30] and Brantut et al.[31] and references therein. This model does not include the possible effects of deformation (beyond poroelastic effects), compaction or shear-induced dilatancy. Advective heat flow is not included as it has been shown to be negligible at a permeability < $10^{-16}$ m$^2$ (ref. [30]). Following Brantut et al.[31], we do not consider the possibility of fluid overpressure at the fault interface leading to tensile failure. In this model we assume that the magnetite layer does not undergo any phase transformation and represents a fully impermeable fluid boundary. Initial pore pressure is set to 0.3 $\sigma_N$ and initial permeability of $10^{-19}$ m$^2$ is taken from laboratory measurements of the permeability of foliated serpentinite[43,47,64].

TEM (Fig. 2) and Raman spectroscopy[37] observations show that the serpentinite within the Livingstone Fault is dominated by chrysotile and lizardite, and that antigorite was not stable. Based on the presence of chrysotile and lizardite and lack of stability of antigorite, we estimate the ambient temperature to be ~300–350 °C. From this, we set the initial temperature in the model ($T_0$) to be 350 °C.

**Model configuration**. The model consists of a 300 µm-thick layer of magnetite between layers of serpentinite (Supplementary Figure 1). The model has four boundaries (B1, B2, B3 and B4). The two external boundaries (B1 and B4) are set to a zero flux condition (insulating boundaries). These boundaries are far enough away from the heat source so as to not induce boundary effects. B3 is a conductive boundary. B2 is a boundary heat source that represents the frictional heating at the fault plane between one side of the magnetite layer and the adjacent serpentinite. This geometry approximates the bimaterial fault interface observed in the natural polished fault surfaces. The mesh maximum element size was set to $1 \times 10^{-6}$ m. The maximum element growth rate, which limits the maximum increase in size

between two adjacent elements, was set to 10%. The resolution of narrow regions is set to 20. The 'resolution of narrow regions' parameter is an arbitrary nonnegative scalar that controls the number of mesh elements that are created in narrow regions of the mesh, with a higher value resulting in a finer mesh in narrow regions.

**1D transport of heat**. The 1D transfer of heat is modelled through the Heat Transfer Module in Comsol. This is governed by the heat equation:

$$\rho C_p \frac{\partial T}{\partial t} = \frac{k}{\rho_s C_p} \frac{\partial^2 T}{\partial x^2} + Q_f + Q_d \qquad (4)$$

where $\rho$, $C_p$ and $k$ are respectively the density, specific heat capacity and thermal conductivity of the fault rocks (serpentinite on one side of the fault, and magnetite and serpentinite on the other). $Q_f$ is the frictional heat flux defined in eq. 5 and $Q_d$ is the latent heat of dehydration of serpentinite.

**Frictional heat flux**. The frictional heat flux from the primary slip surface ($Q_f$) set at boundary B2 is defined as:

$$Q_f = \mu(\sigma_N - p_f)f(t) \qquad (5)$$

where the coefficient of friction $\mu = 0.4$, the normal stress $\sigma_N = 270$ MPa, the pore pressure $p_f$ is initially set to $p_f = 0.3\sigma_N$ and the slip velocity function $f(t)$ corresponds to a boxcar function (smoothed square-wave pulse) with amplitude of 1 m/s. for the slip duration $d/v$, where $d$ is the fault displacement and $v$ is the velocity.

When coseismic frictional heating occurs, initially present pore fluids and fluids produced during dehydration reactions expand, which reduces the effective normal stress and the frictional heating according to eq. 5.

**Comparison of numerical model with analytical solution**. Before adding the coupled thermal processes (thermal pressurisation and coseismic dehydration) to

the numerical model, we first compare the basic numerical model (considering only the 1D transport of heat produced at the slip surface) to the analytical solution of McKenzie and Brune[48]:

$$T(x,t) = \frac{\sigma_f}{2\rho C_p \sqrt{K\pi}} \int_0^t e^{\left[-\frac{x^2}{4K(t-t_0)}\right]} \frac{d}{t_1\sqrt{t-t_0}} dt_0 \tag{6}$$

For the derivation of the analytical solution and definition of the parameters, see McKenzie and Brune[48]. In this case, we consider a zero-width fault located at $x = 0$ and a displacement of $d = 0.01$ m. For simplicity, the rock consists of magnetite on either side of the fault plane. The resulting solutions from the analytical model and the numerical model agree to within < 1% (Supplementary Figure 2).

**Enthalpy of dehydration of serpentinite.** Accounting for the enthalpy of dehydration of serpentinite is done through the Phase Change Material node in the Heat Transfer Module in Comsol. While a more rigorous treatment of coseismic dehydration reactions can be achieved by describing the dehydration reaction with the Arrhenius equation and assuming an order of reaction in the rate eq.[31], there remains uncertainty in choice of reaction order and kinetic parameters. It is also important to consider that during coseismic frictional heating, reactions are likely driven by kinetics and reaction boundaries can be significantly overstepped.

We model the dehydration of serpentinite by considering it to be a phase change using the COMSOL Phase Change Material node in the Heat Transfer Module. As an approximation, we consider the dehydration reaction as a phase change from serpentine to olivine (forsterite). The Phase Change Material node uses the apparent heat capacity formulation to model this transition[65]. The enthalpy of dehydration is added to the heat transfer equation (eq. 4) over an interval between $T_d - \frac{\Delta T}{2}$ and $T_d + \frac{\Delta T}{2}$, where $T_d$ is the dehydration temperature (550 °C) and $\Delta T$ is set to 100 °C. Within this interval, the fault rock properties are modulated by a smoothed function $\xi$ that represent the progress of the reaction, which is equal to 0 before the reaction and 1 after the reaction. The effective thermal properties of the fault rock in eq. 4 during coseismic dehydration are modelled such that:

$$\rho_{\text{eff}} = (1 - \xi)\rho_{\text{serpentinite}} + \xi\rho_{\text{forsterite}} \tag{7}$$

$$k_{\text{eff}} = (1 - \xi)k_{\text{serpentinite}} + \xi k_{\text{forsterite}} \tag{8}$$

$$C_{p_{\text{eff}}} = \frac{1}{\rho_{\text{eff}}}\left((1 - \xi)\rho_s C_{p_{\text{serpentinite}}} + \xi\rho_s C_{p_{\text{forsterite}}}\right) + C_L \tag{9}$$

where $\rho_{\text{eff}}$, $k_{\text{eff}}$ and $C_{p_{\text{eff}}}$ are respectively the effective density, thermal conductivity and heat capacity. $C_L$ is the latent heat distribution. For additional information, refer to the Heat Transfer Module user guide in the COMSOL documentation.

**Thermal pressurisation in a poroelastic medium.** Pore fluid pressure is calculated through the fluid mass conservation equation, which takes into account the pressurisation of the fluid by heating, the fluid flow away from the fault, the poroelasticity and the fluids and porosity produced by the dehydration reaction. The energy equation (eq. 4) is coupled to the fluid mass conservation equation through the pore pressure parameter $p_f$.

$$\frac{\partial P}{\partial t} = \left(\frac{\lambda_f - \lambda_n}{\beta_f + \beta_n}\right)\frac{\partial T}{\partial t} + \frac{1}{\rho_w n_0 (\beta_f + \beta_n)}\frac{\partial P}{\partial x}\left(\rho_w \frac{K_s}{\eta_w}\frac{\partial P}{\partial x}\right)$$
$$+ \left(\frac{\rho_{\text{serpentinite}}}{\rho_w}w_{\text{serpentine}}\frac{\nu_s M_{H_2O}}{M_{\text{serpentine}}} - \Delta n\right)\frac{1}{n_0 (\beta_f + \beta_n)}\frac{\partial \xi}{\partial t} \tag{10}$$

For a complete derivation of this equation, see Brantut et al.[31]. All parameters are listed and described in Supplementary Table 2. This equation is evaluated through the PDE interface in the Mathematics node in COMSOL.

**Evolution of permeability.** The dehydration of serpentinite leads to formation of reaction-enhanced porosity. This creates a porosity front that advances with the progress of serpentinite dehydration. The increase in porosity ($\Delta n$ in eq. 10) results in an increase in local fluid storage capacity, diminishing the effect of thermal pressurisation. In addition, Tenthorey and Cox[43] found that dehydration of serpentinite creates transiently interconnected porosity, which in turn results in a local increase in permeability. Based on the data of Tenthorey and Cox[43], we allow the permeability to increase by up to an order of magnitude within the reaction zone in relation to the reaction progress function $\xi$:

$$K_{rxn} = K_s(1 + 10\xi)$$

## Data availability

All data and values of parameters used in this study are available in the paper and the Supplementary Information (and references therein). Any additional data that support the findings of this study are available from the corresponding author upon request.

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

## Acknowledgements

This work was supported by the Marsden Fund Council (project UOO1417 to Smith) administered by the Royal Society Te Apārangi, with additional funding from a University of Otago Research Grant. We thank Marianne Negrini, Brent Pooley, Claudia Magrini and Giovanna Giorgetti for technical support.

## Author contributions

M.S.T. and S.A.F.S. carried out fieldwork and performed microstructural analysis of fault rocks. M.S.T. and C.V. performed transmission electron microscopy. M.S.T. performed numerical modelling with input from S.A.F.S., M.S.T. wrote the manuscript with discussion and input from all authors. S.A.F.S. and J.M.S. supervised the whole project.

## Additional information

**Competing interests:** The authors declare no competing interests.

