## [Peer Review File · Nature Communications]

Reviewers' comments:

Reviewer #1 (Remarks to the Author):

The paper by Tarling et al. presents field evidence for co-seismic serpentinite amorphisation and dehydration. These new observations confirm what has been inferred theoretically and experimentally regarding the potential seismogenic nature of serpentinite (or at least their ability to propagate rupture, if not nucleate it). The authors combine their nanostructural observations to provide a temperature profile across a fault zone during an earthquake, which is a very elegant application and further confirms their co-seismic hypothesis.

I am not a petrologist so I cannot make deep comments about the authors' petrological observations, although they appear to be solid and well documented.

I have one major comment regarding the modelling. The method is not described in any detail even in the supplementary materials. It seems that the authors perform a 2D computation, but this is completely unnecessary because the problem is only 1D, across the fault. Furthermore, the assumption of a friction coefficient of 0.4 is not justified. In laboratory experiments, the dry friction coefficient varies strongly with slip and slip rate, and is likely well below 0.4 at coseismic slip rates. In addition, the authors perform "dry" simulations and do not account for the effects of fluids, even though they mention that this effect is important in the main text. Based on their microstructural data, the author could estimate the fluid volume released by antigorite amorphisation and dehydration (per unit fault surface), estimate the potential pore pressure rise and check whether the assumption of dry friction is a good one (likely not). Also, the input slip velocity is simply a constant equal to 1 m/s, which is not realistic. Taken together, the assumptions (dry friction equal to 0.4, no fluids, constant high slip rate, zero-width shear zone) tend to produce a massive overestimation of the heat production and temperature rise. So their estimate of Mw around 2 to 3 is likely extremely low. Overall, I find the modelling part not very convincing and it does not seem to bring much to the story, unless it is taken to another level with better justifications for the parameters used, and including the effects of fluids and realistic slip rate evolution.

I also have another comment about the temperature estimates based on the observations. It seems that the authors base these estimations on equilibrium reaction boundaries (most of the references in table s4 are from petrological studies). But during coseismic slip the reactions are likely more driven by kinetics, and reaction boundaries can be overstepped significantly. This is not a problem this paper can solve, but it must be discussed and the authors should mention whether their estimated temperature profile is a lower or an upper bound. The use of error bars in their figure 2 is highly misleading because these error do not correspond to statistics on multiple measurements at the same distance from the sliding surface, but to a standard deviation from petrological studies.

Reviewer #2 (Remarks to the Author):

Review of "Dynamic earthquake rupture preserved in a creeping serpentinite"

Summary: The authors document detailed microstructural products associated with slip surfaces in a serpentinite/magnetite shear zone that – when examined in the context of their spatial distribution, allow for the reconstruction of a thermal impulse associated with rapid slip. These microstructural fingerprints that grade from magnetite with no evidence of melting on the slip surface to aggregates of nanocrystalline olivine and enstatite a few tens of microns from the slip surface to partial amorphization of serpentinite minerals less than 200 microns from the slip surface. The authors

construct numerical models using COMSOL Multiphysics to provide upper and lower bounds for seismic slip events that produce thermal profiles satisfying the petrological/microstructural constraints, leading them to conclude that the observations can be explained by Mw2-2.7 earthquakes that propagated through the serpentinite shear zones. This is a very well-written manuscript that represents an important field analog for laboratory and theoretical investigations in serpentinite-rich shear zones, and, as such merits serious consideration for publication. I do have some concerns about the modeling aspect of the paper, and I would expect some substantial clarification before publication.

Main Comments:

Most of my comments are related to the modeling of field observations: In general, much more information needs to be reported about the modeling approach. COMSOL is a wonderful tool, but it can be dangerous as a black box, and while the authors all have a strong track records, I think the bulk of their collective expertise is in other techniques, but there are a number of things that need to be reported for such models to verify that assumptions are reasonable and the results could be repeated:

1. What is the form of the governing equations you are solving (in the strong form, not the weak form)? It is difficult to evaluate the assumptions and the relative importance of the various constants, independent and dependent variables in the model without this information. One of the challenges with COMSOL is that sometimes it can be so easy to use that the user is not aware of the assumptions built into the model. Furthermore, for someone trying to reproduce the results and analyze the results this is absolutely an essential thing to report. Some of the points below (particularly point 5) would likely be at least partially answered by reporting the governing equations.
2. What were the boundary and initial conditions along the lower, upper, and lateral boundaries? I infer from the estimate of the ambient temperature that the lower and upper boundaries the initial and boundary conditions are both a fixed temperature, but what about the lateral boundaries? Are the boundary conditions there stated as a constant value or a flux?
3. Since this is a 2D model, where are the temperature profiles shown in Figure 2 evaluated? I find it surprising that the problem is solved in a 2D domain when it is really a 1D problem.
4. There are several coupled processes in this model – (presumably) conductive heat flow, temperature-dependent density, thermal conductivity, and heat capacity, as well as phase transformations. Before coupling the thermal processes, do the numerical simulations match the predictions from analytical solutions? The more complex the model, the more places there are for hidden errors, so it is essential to validate at each step of added complexity.
5. Are there any other coupled processes included in the model? For example, is the medium rigid or deformable? Fluid flow and/or Poroelastic effects? Advective heat flow? My guess is that any phase transition would be accompanied by volume change. Also, the authors state in Lines 137-139 that “The discovery of high temperature dehydration assemblages indicates that significant volumes of H₂O can be released during coseismic slip in serpentinite, suggesting that thermal pressurization is a likely dynamic weakening mechanism in natural serpentinite shear zones”. This certainly may explain why no melt products are found, but if so, how might this process (release of water) affect the results of your simulations? Might the ruptures have actually been much smaller? Note that I don’t necessarily advocate for incorporating all of this complexity into the model (simpler is often much more instructive), but all assumptions should be stated up front (in this case, probably in the SI), but the potential implications of major assumptions (like release of water, slip rate) that are not explicitly investigated need to be discussed explicitly.

Line-by-line comments:

Lines 28-29: “Our results indicate creeping shear zones... do not represent barriers to earthquake rupture.” This is a bit of an overstatement. While not completely wrong, the black-and-white nature of the sentence does not accurately reflect the nuance of the implication. Other works, including Noda and Lapusta (2013) – citation #7 and French et al. (2014) point out the fact that the rate-dependent

frictional response of rock materials is itself rate dependent. Large enough velocities or rapid accelerations at the leading tip of an earthquake rupture can make a material that is velocity strengthening at low slip rates velocity weakening. Therefore, a fault patch that is a barrier to seismic slip on shorter time intervals may actually rupture at longer intervals. While I find the results of this manuscript interpreting products of seismic slip convincing, there is certainly no evidence that I see that suggests these rocks did not present a barrier to seismic slip.

Line 123-125: In my opinion, the "scaling laws" here merit a reference

Line 133: I think this should be shear zone(s) – plural

Line 152: Note that nucleation and propagation of earthquakes are two different things (they seem to be interchanged in the manuscript without a clear distinction), and I don't think that evidence that earthquakes nucleated in these pods is very convincing. Is there any evidence that earthquakes nucleated here beyond the existence of material heterogeneity? Most experimental and theoretical evidence I am aware of where unstable slip occurs in otherwise rate-strengthening materials (see Noda & Lapusta and French et al. reference above) indicate that this behavior only happens at very high slip speeds and/or under imposed rapid accelerations. Could the observations in this paper not simply be explained by an earthquake rupture propagating into this patch and inducing velocity weakening deformation (even if it is distributed across multiple slip surfaces)? To me this is a much more convincing scenario. Including uncertainties related to inherent slip – and slip rate – heterogeneity across faults during single earthquake ruptures (e.g., Rowe and Griffith, 2015), the potential presence of water/fluid and thermal pressurization as described above in the modeling comments, and slip distributed among multiple slip surfaces, my guess is that the range of earthquake magnitudes concluded in this paper are likely a lower bound on the actual size.

Supplementary Information, Line 99: "dependant" should be spelled "dependent"

References:

- French, M. E., Kitajima, H., Chester, J. S., Chester, F. M., & Hirose, T. (2014). Displacement and dynamic weakening processes in smectite-rich gouge from the Central Deforming Zone of the San Andreas Fault. *Journal of Geophysical Research: Solid Earth*, 119(3), 1777-1802.
- Rowe, Christie D., and W. Ashley Griffith. "Do faults preserve a record of seismic slip: A second opinion." *Journal of Structural Geology* 78 (2015): 1-26.

Response to Reviewers' comments:

Thank you for the opportunity to address the weaknesses in our manuscript and improve our numerical modelling approach. Based on the review comments, we have made major revisions to the numerical model. This has essentially involved starting the numerical model from scratch, adopting a 1D approach, and progressively implementing and testing a range of more realistic parameters. Significantly, in implementing many of the changes suggested by the reviewers, *we have been able to produce a numerical model that fits our petrological data much better than the original model, and provides additional insights in to key earthquake rupture processes.* In this regard, the reviews have proven to be extremely constructive, and we would like to thank the reviewers for their critical assessments.

Before providing a point-by-point response to the reviewers' comments, we will briefly outline the main changes implemented in the revised paper:

1. The numerical modelling is now performed in 1D.
2. The numerical model has been completely updated to include the effects of thermal pressurisation, fluid production, and volume changes (i.e. porosity increase) during coseismic dehydration of serpentinite.
3. The thermal profiles predicted by the new model (new Figure 3) suggest an earthquake magnitude of between $M_w=2.7-4.0$. The shape of the new thermal profiles fits our petrological data much better than the previous model.
4. We have completely rewritten the supplementary materials to include a full description of the new model, including the most important assumptions, the relevant governing equations, and our adopted values for all parameters.
5. Parts of the main text have been changed to account for the reviewers' comments. Much of the second half of the paper has been rewritten to account for the new numerical model.

Below we include a point-by-point response to the reviewers' comments. Changes to the text are also highlighted by a different colour in the revised manuscript.

Reviewer comments are in **bold**, author responses are in *italics*.

Responses to Reviewer 1

I have one major comment regarding the modelling. The method is not described in any detail even in the supplementary materials.

We have rewritten the supplementary materials to include a complete description of the new modelling approach. Our modelling is now based mainly on the mathematical treatment of thermal

pressurisation and coseismic dehydration outlined by Rice (2006) and furthered by Brantut et al. (2010). We have included all of the relevant governing equations and appropriate citations to the proofs of the governing equations.

It seems that the authors perform a 2D computation, but this is completely unnecessary because the problem is only 1D, across the fault.

Based on this comment and a similar comment from reviewer 2, we reconstructed the model in 1D. All of our numerical modelling in the revised paper was performed in 1D. We thank the reviewer for this comment because a 1D approach is much more appropriate for this set of problems, and the increased efficiency of modelling in 1D also allowed us to more fully explore a wide range of parameter space.

Furthermore, the assumption of a friction coefficient of 0.4 is not justified. In laboratory experiments, the dry friction coefficient varies strongly with slip and slip rate, and is likely well below 0.4 at coseismic slip rates.

We agree that a single friction coefficient of 0.4 is probably not realistic, and that at high slip rates the friction coefficient is likely to be lower due to dynamic weakening. However, our new model now captures some of the effects of the dynamic weakening process by including thermal pressurisation. Instead of modelling a slip- or slip-rate dependent friction coefficient, we allow thermal pressurisation to decrease the effective normal stress and the resultant frictional heating. We also note that previous models of earthquake slip have adopted a fixed value of the friction coefficient, because there is currently no consensus on how to properly implement the evolution of slip- and slip rate-dependent friction.

In addition, the authors perform "dry" simulations and do not account for the effects of fluids, even though they mention that this effect is important in the main text. Based on their microstructural data, the author could estimate the fluid volume released by antigorite amorphisation and dehydration (per unit fault surface), estimate the potential pore pressure rise and check whether the assumption of dry friction is a good one (likely not).

This is an excellent point and we fully agree that our "dry" modelling approach was not appropriate. Based on this and other comments, we have updated and refined our new model to include the effects of fluids; both the initial pore fluids, and the fluids that are generated during the dehydration of serpentine. Rather than estimating the possible fluid volume produced (which is difficult since the maximum temperature and extent of dehydration are influenced by the production and thermal pressurisation of the fluids themselves), we have applied the mathematical treatment of thermal pressurization established by Rice (2006) and Brantut et al. (2010). The assumption of dry friction was indeed inappropriate. The initial presence of pore fluids leads to significant thermal pressurisation, which, when combined with fluids produced by the dehydration of serpentine, results in a significant decrease in coseismic heat production. However, this decrease in coseismic heat production is partly offset by the formation of reaction-enhanced porosity, which allows partial draining of the fluids – this is discussed in the manuscript at lines 115-132 and in the supplementary information in section 2.6.

Also, the input slip velocity is simply a constant equal to 1 m/s, which is not realistic.

We agree that a realistic and appropriate source time function is critical when performing earthquake kinematic source inversions. However, for the purpose of this numerical model, we believe that the area under the curve of the slip velocity function (SVF) is the most important characteristic, rather than the shape of the SVF, since the area determines the overall heating that occurs. To illustrate this point, the plots below show the output of our numerical model considering a boxcar SVF (red curve; a smoothed square-function, as used in our current and previous model configuration) and a truncated modified Yoffe function (MYF) (blue curve; consistent with 'pulse-like' nature earthquake ruptures). For the MYF, we applied the formulation of Tinti et al. (2005) with the corrections of Bizzarri (2012). We apply the same ratios of the free parameters of the MYF (t_r and t_s) as Bizzarri (2012). We truncate the MYF such that the duration is equal for both SVFs. The results show that the area below the curves of both SVFs is approximately equal, which means that the heat production is also approximately equal.

Taken together, the assumptions (dry friction equal to 0.4, no fluids, constant high slip rate, zero-

width shear zone) tend to produce a massive overestimation of the heat production and temperature rise. So their estimate of Mw around 2 to 3 is likely extremely low.

Our updated numerical model now takes into account the effects of thermal pressurisation of pore fluids, and the production of fluids released by serpentine dehydration. We currently lack constraints on the coseismic slip zone width and the thickness of dehydrated serpentine adjacent to the magnetite, which is why we use a zero-width slip zone. Although this results in an overestimation of heat production at the fault surface for a given slip distance, it does provide a minimum slip distance required to reproduce a given thermal profile. This is an important point that we state clearly in the revised manuscript at lines 154-157.

As suggested by the reviewer, the original estimates of earthquake magnitude were too low. Our new model including the effects of fluids now predicts an earthquake magnitude range between 2.7 and 4.0.

Overall, I find the modelling part not very convincing and it does not seem to bring much to the story, unless it is taken to another level with better justifications for the parameters used, and including the effects of fluids and realistic slip rate evolution.

We are extremely thankful for this critical analysis of our modelling. The updated numerical model - including the effects of fluids through the treatment of thermal pressurisation and fluid production during serpentine dehydration - results in a much better fit to our petrological data compared to our original model. The review comments have also motivated us to strengthen the supplementary information by including governing equations and information regarding justification of our chosen parameters.

I also have another comment about the temperature estimates based on the observations. It seems that the authors base these estimations on equilibrium reaction boundaries (most of the references in table s4 are from petrological studies). But during coseismic slip the reactions are likely more driven by kinetics, and reaction boundaries can be overstepped significantly. This is not a problem this paper can solve, but it must be discussed and the authors should mention whether their estimated temperature profile is a lower or an upper bound.

This is an excellent point that we neglected to address in the original manuscript. We have updated the manuscript at lines 106-111 to address and briefly discuss the implications of this with regard to the possibility of overstepped reaction boundaries. Based on the choices of parameters (e.g fault width) and the possibility of overstepping reaction boundaries, our numerical model could represent a lower bound in terms of possible earthquake magnitude responsible for the coseismic dehydration. We have also made this clear at lines 197-200 in the revised manuscript.

The use of error bars in their figure 2 is highly misleading because these error do not correspond to statistics on multiple measurements at the same distance from the sliding surface, but to a standard deviation from petrological studies.

We agree with this and believe that the bars on the previous figure 2 were misleading. To clarify the meaning of these bars (on revised Figure 3), we have added an inset to Figure 3 that shows what x and y bars correspond to. The x-axis error bars correspond to the standard error on the mean with a

95% confidence interval, of data taken from an extensive literature review of petrological studies dealing with dehydration reactions (Table S1). The y-axis bars correspond simply to the range of distances that each dehydration product was found in our SEM and TEM analysis.

Responses to Reviewer 2

Most of my comments are related to the modeling of field observations: In general, much more information needs to be reported about the modeling approach. COMSOL is a wonderful tool, but it can be dangerous as a black box, and while the authors all have a strong track records, I think the bulk of their collective expertise is in other techniques, but there are a number of things that need to be reported for such models to verify that assumptions are reasonable and the results could be repeated:

1. What is the form of the governing equations you are solving (in the strong form, not the weak form)? It is difficult to evaluate the assumptions and the relative importance of the various constants, independent and dependent variables in the model without this information. One of the challenges with COMSOL is that sometimes it can be so easy to use that the user is not aware of the assumptions built into the model. Furthermore, for someone trying to reproduce the results and analyze the results this is absolutely an essential thin to report. Some of the points below (particularly point 5) would likely be at least partially answered by reporting the governing equations.

Thank you for bringing these issues to our attention, and we agree that our presentation of our modelling approach was insufficient in the original article. As outlined in our responses to reviewer 1, we have completely updated our numerical model based on the review comments. We now report all governing equations and assumptions in the supplementary materials, and discuss potential limitations of the model in the main manuscript.

2. What were the boundary and initial conditions along the lower, upper, and lateral boundaries? I infer from the estimate of the ambient temperature that the lower and upper boundaries the initial and boundary conditions are both a fixed temperature, but what about the lateral boundaries? Are the boundary conditions there stated as a constant value or a flux?

We have updated the supplementary materials to clearly state all of the initial boundary conditions, and have included a new supplementary figure (Fig S1) to clarify the model geometry. The new model, now evaluated in 1D, has 2 external boundaries at which a zero flux condition is set (insulating boundaries). However, we have been careful to set up these boundaries at a large enough distance from the heat source boundary (B2 in the figure) that they do not influence the thermal evolution of the model.

3. Since this is a 2D model, where are the temperature profiles shown in Figure 2 evaluated? I find it surprising that the problem is solved in a 2D domain when it is really a 1D problem.

We thank the reviewer for this comment. We have completely updated the model based on this and other comments and now perform all calculations with a 1D model.

4. There are several coupled processes in this model – (presumably) conductive heat flow, temperature-dependent density, thermal conductivity, and heat capacity, as well as phase transformations. Before coupling the thermal processes, do the numerical simulations match the predictions from analytical solutions? The more complex the model, the more places there are for hidden errors, so it is essential to validate at each step of added complexity.

This is a very good point, and we agree that the original article did not show sufficient validation of the model set up and initial outputs. To improve this, we have added a comparison in the supplementary materials (Fig. S2) to show how our basic numerical model compares with the widely-accepted analytical solution of McKenzie & Brune (1972). The results show that our numerical model matches predictions of the analytical solution (McKenzie and Brune 1972) to within 1%, before we start to add the effects of coupled thermal processes. This gives additional confidence that our model set up is satisfactory.

5. Are there any other coupled processes included in the model? For example, is the medium rigid or deformable? Fluid flow and/or Poroelastic effects? Advective heat flow? My guess is that any phase transition would be accompanied by volume change. Also, the authors state in Lines 137-139 that “The discovery of high temperature dehydration assemblages indicates that significant volumes of H₂O can be released during coseismic slip in serpentinite, suggesting that thermal pressurization is a likely dynamic weakening mechanism in natural serpentinite shear zones”. This certainly may explain why no melt products are found, but if so, how might this process (release of water) affect the results of your simulations? Might the ruptures have actually been much smaller? Note that I don’t necessarily advocate for incorporating all of this complexity into the model (simpler is often much more instructive), but all assumptions should be stated up front (in this case, probably in the SI), but the potential implications of major assumptions (like release of water, slip rate) that are not explicitly investigated need to be discussed explicitly.

This is an excellent comment and similar to those made by reviewer 1 on the lack of fluid effects in our original modelling approach. As described in our replies to reviewer 1, our new numerical model includes the effects of pore fluid pressurization, as well as thermal pressurization during dehydration of serpentine. We have updated the manuscript and the supplementary materials to detail all of the coupled processes that are included in the new model, including the effects of fluids. Additionally, the assumptions are now listed in the supplementary materials. In the updated model, the medium is rigid and we do not account for deformation. Fluid flow and poroelastic effects are included in the thermal pressurisation equation (Rice 2006). The phase transition from serpentine to olivine is indeed accompanied by a net volume change that generates ~24% porosity, which plays an important role in accommodating the fluid pressure produced by co-seismic dehydration (Brantut et al. 2010). The overall effect of including these coupled processes in the new model is a thermal profile that more closely matches our petrological data, and a revised range of earthquake magnitudes. The larger earthquake magnitudes predicted by our new model arise from the effects of thermal pressurisation and coseismic dehydration, which inhibit temperature rise in the fault (Brantut et al. 2010, 2011)

Line-by-line comments:

Lines 28-29: “Our results indicate creeping shear zones... do not represent barriers to earthquake rupture.” This is a bit of an overstatement. While not completely wrong, the black-and-white nature of the sentence does not accurately reflect the nuance of the implication. Other works, including Noda and Lapusta (2013) – citation #7 and French et al. (2014) point out the fact that the rate-dependent frictional response of rock materials is itself rate dependent. Large enough velocities or rapid accelerations at the leading tip of an earthquake rupture can make a material that is velocity strengthening at low slip rates velocity weakening. Therefore, a fault patch that is a barrier to seismic slip on shorter time intervals may actually rupture at longer intervals. While I find the results of this manuscript interpreting products of seismic slip convincing, there is certainly no evidence that I see that suggests these rocks did not present a barrier to seismic slip.

We agree with the reviewer that the previous sentence was an oversimplification and was not justified based on our observations. We have revised this sentence to more closely reflect the main implications of our results, namely that we have good evidence for propagation of a dynamic earthquake rupture in serpentinite, but not necessarily the nucleation of a rupture.

Line 123-125: In my opinion, the “scaling laws” here merit a reference

We have added a reference to address this issue (Sibson 2011).

Line 133: I think this should be shear zone(s) – plural

The suggested correction has been made.

Line 152: Note that nucleation and propagation of earthquakes are two different things (they seem to be interchanged in the manuscript without a clear distinction), and I don’t think that evidence that earthquakes nucleated in these pods is very convincing. Is there any evidence that earthquakes nucleated here beyond the existence of material heterogeneity? Most experimental and theoretical evidence I am aware of where unstable slip occurs in otherwise rate-strengthening materials (see Noda & Lapusta and French et al. reference above) indicate that this behavior only happens at very high slip speeds and/or under imposed rapid accelerations. Could the observations in this paper not simply be explained by an earthquake rupture propagating into this patch and inducing velocity weakening deformation (even if it is distributed across multiple slip surfaces)? To me this is a much more convincing scenario. Including uncertainties related to inherent slip – and slip rate - heterogeneity across faults during single earthquake ruptures (e.g., Rowe and Griffith, 2015), the potential presence of water/fluid and thermal pressurization as described above in the modeling comments, and slip distributed among multiple slip surfaces, my guess is that the range of earthquake magnitudes concluded in this paper are likely a lower bound on the actual size.

We fully agree with the reviewer that nucleation and propagation of earthquakes are two distinct processes, and that such a distinction was not sufficiently clear in the original manuscript. In addition to this, we also feel that we included some sentences in our original manuscript that were not strongly supported by our field and microstructural observations, and that such sentences confused the issue of earthquake nucleation vs. propagation. In the revised manuscript, we have carefully reworded a number of sentences so that the focus of the manuscript is on our strongest

conclusion: namely, the petrological evidence, and support from numerical modelling, for the preservation of a dynamic earthquake rupture in serpentinite. We agree with the reviewer that we do not have good observations to say where the dynamic rupture nucleated. To reflect this, we have removed the final sentence of the abstract that was speculative – the abstract now ends with our strongest conclusion regarding the petrological and microstructural evidence.

Given the assumptions and limitations with numerical models in general, including the new numerical model in our paper, we agree that our suggested range of earthquake magnitudes represents a lower bound. We have revised the manuscript to more clearly state this point (lines 197-200).

Supplementary Information, Line 99: “dependant” should be spelled “dependent”

The suggested correction has been made.

References

1. Rice, J. R. Heating and weakening of faults during earthquake slip. *J. Geophys. Res. Solid Earth* **111**, 1–29 (2006).
2. Brantut, N., Schubnel, A., Corvisier, J. & Sarout, J. Thermochemical pressurization of faults during coseismic slip. *J. Geophys. Res. Solid Earth* **115**, 1–17 (2010).
3. Noda, H. & Shimamoto, T. Thermal pressurization and slip-weakening distance of a fault: An example of the Hanaore fault, southwest Japan. *Bull. Seismol. Soc. Am.* **95**, 1224–1233 (2005).
4. Tinti, E., Fukuyama, E., Piatanesi, A. & Cocco, M. A kinematic source-time function compatible with earthquake dynamics. *Bull. Seismol. Soc. Am.* **95**, 1211–1223 (2005).
5. Bizzarri, A. Analytical representation of the fault slip velocity from spontaneous dynamic earthquake models. *J. Geophys. Res. Solid Earth* **117**, 1–21 (2012).
6. McKenzie, D. & Brune, J. N. Melting on Fault Planes During Large Earthquakes. *Geophys. J. R. Astron. Soc.* **29**, 65–78 (1972).
7. Brantut, N., Han, R., Shimamoto, T., Findling, N. & Schubnel, A. Fast slip with inhibited temperature rise due to mineral dehydration: Evidence from experiments on gypsum. *Geology* **39**, 59–62 (2011).
8. Sibson, R. H. The scope of earthquake geology. *Geol. Soc. London, Spec. Publ.* **359**, 319–331 (2011).

REVIEWERS' COMMENTS:

Reviewer #1 (Remarks to the Author):

The authors did modify the manuscript according to all my and the other reviewer's comments, and they addressed all the main points satisfactorily.

I have no additional remarks and I suggest this paper to be published in its present form.

Reviewer #2 (Remarks to the Author):

Summary:

This is a revised manuscript documenting strong evidence for paleo-earthquakes in exhumed serpentinite fault zone. The revisions, which focus primarily on the modeling approach and presentation of model results, have sufficiently addressed my previous criticisms. The result a very well-written manuscript that represents an important field analog for laboratory and theoretical investigations in serpentinite-rich shear zones. As it is the first such report, it will likely be a highly cited paper that should be published in Nature Communications. I do have two very minor suggestions for the authors before recommending the paper be formally accepted.

Main Text:

Line 142: Technically, this term Q has units of $\text{Pa}\cdot\text{m}/\text{s}$ which is equivalent to W/m^2 . Therefore, rather than simply "Frictional heat", this should be referred to as "Frictional heat flux from the primary slip surface". Also, it is usually preferable to use the present tense when referring to work done in the presented study.

Supplementary material:

Line 40: It still isn't clear what a maximum element growth rate of 1.1 (units?) and resolution of narrow regions of 20 (units?) means. Please clarify.

Below we include a point-by-point response to the reviewers' comments.

Reviewer comments are in **bold**, author responses are in *italics*.

Responses to Reviewer 2

Main Text:

Line 142: Technically, this term Q has units of $\text{Pa}\cdot\text{m}/\text{s}$ which is equivalent to W/m^2 . Therefore, rather than simply "Frictional heat", this should be referred to as "Frictional heat flux from the primary slip surface". Also, it is usually preferable to use the present tense when referring to work done in the presented study.

We have revised the text to refer to the term Q as the "frictional heat flux from the primary slip surface" throughout. We have also altered the text so that the present tense is used when referring to the modelling work that was performed.

Supplementary material:

Line 40: It still isn't clear what a maximum element growth rate of 1.1 (units?) and resolution of narrow regions of 20 (units?) means. Please clarify.

We have added a more detailed explanation for the significance of these parameters.